# Position: Compatibility-First Design Is Critical for Progress in Agentic Memory

## Abstract

This position paper argues that neither uncoordinated fragmentation nor rigid standardisation is desirable for agentic memory systems in machine learning research. Instead, we advocate for compatibility-first design: shared interfaces and minimal common abstractions that enable heterogeneous memory systems to interoperate without constraining internal architectures. Fragmentation leads to wasted effort, harder-to-maintain code, difficulty comparing results, and increased safety risks, while premature standardisation risks hard-coding decisions in a rapidly evolving space. Compatibility at the interface level captures the benefits of coordination: reduced redundancy plus shared benchmarks and safety practices, without imposing a monolithic standard. We examine credible alternative viewpoints favouring fragmentation or early standardisation, discuss their limitations, and identify concrete steps, including community-driven interface specifications, reference implementations, and interoperable benchmarks to realise the compatibility-first agenda.

## 1. Introduction

Agentic memory systems (Dilocker et al., 2019; Pinecone, 2019; Team, 2022; ElizaOS, 2023; Packer et al., 2023; OpenAI, 2024; Team, 2024; VolcEngine, 2025; MemoDB, 2025; Kang et al., 2025; Wang & Chen, 2025; LangChainAI, 2025; SuperMemoryAI, 2025; CogneeAI, 2025; Chhikara et al., 2025; Rasmussen et al., 2025; Xu et al., 2025; Li et al., 2025; Memobase, 2025; Zhang et al., 2025a; MemoriLabs, 2025; Cao et al., 2025; Wei et al., 2025a; MemU, 2025; Latimer et al., 2025; Hu et al., 2026) have become a central component of modern Artificial Intelligence (AI) agents, enabling persistent context, long-horizon reasoning, and adaptive behaviour across interactions. These capabilities are critical for applications such as personalised assistants (Yuan et al.,

2025), autonomous decision-making, and multi-agent systems, where agents must retain, update, and reason over information beyond context window limits (Hosseini et al., 2025b). Therefore, recent works focus on efficiency (Bini et al., 2025), expressivity (Hong & He, 2025), and continual learning capabilities (Hosseini et al., 2025a; Yuan et al., 2025; Ocker et al., 2025) of agentic memory architectures.

This rapid progress has been streamlined by specialised benchmarks and evaluation suites (Deng et al., 2023; Maharana et al., 2024; Wu et al., 2024; Zhong et al., 2024; Hsieh et al., 2024; Du et al., 2024; Kuratov et al., 2024; Tan et al., 2025; Miyai et al., 2025; Jiang et al., 2025; Zhang et al., 2025b; Zhao et al., 2025; Jia et al., 2025; Wan & Ma, 2025; He et al., 2025; Zheng et al., 2025; Bai et al., 2025; Wang et al., 2025; Chen et al., 2025; Wu et al., 2025a; Ai et al., 2025; Hu et al., 2025b; Tavakoli et al., 2025; Wei et al., 2025b) that help assess the detailed performance of memory systems and agentic memory architectures. These advances, however, are accompanied by an unprecedented fragmentation (Hu et al., 2025a). Memory systems are often developed as tightly coupled, end-to-end frameworks, with incompatible design assumptions (Wu et al., 2025b).

Research groups and industrial teams frequently reimplement similar core mechanisms, such as storage, retrieval, compression, and update logic, within isolated systems such as LangGraph (LangChain, 2025), LlamaIndex (Liu, 2022), and AutoGen (Microsoft, 2023) among many others (Chase, 2022; CrewAI, 2023; Vercel, 2023; AgnoAI, 2025; CamelAI, 2025; OpenAI, 2025; Google, 2025; MastraAI, 2025). While these frameworks are individually effective, their lack of interoperability results in duplicated effort, growing technical debt, and adds barriers to integration and reuse.

Fragmentation also complicates evaluation, safety, and security (Wibowo & Polyzos, 2025). Siloed development practices hinder coordinated vulnerability discovery, shared audits, and timely patch deployment, while incompatible evaluation protocols make it difficult to meaningfully compare memory-augmented agents across benchmarks or application domains (EverMind-Team, 2025). As a result, reported performance gains are often narrow, task-specific, and difficult to contextualise, limiting their relevance for real-world deployment (Atta et al., 2025). Similar dynamics have been observed historically in other areas of software

[1]Anonymous Institution, Anonymous City, Anonymous Region, Anonymous Country. Correspondence to: Anonymous Author <anon.email@domain.com>.

Preliminary work. Under review by the International Conference on Machine Learning (ICML). Do not distribute.

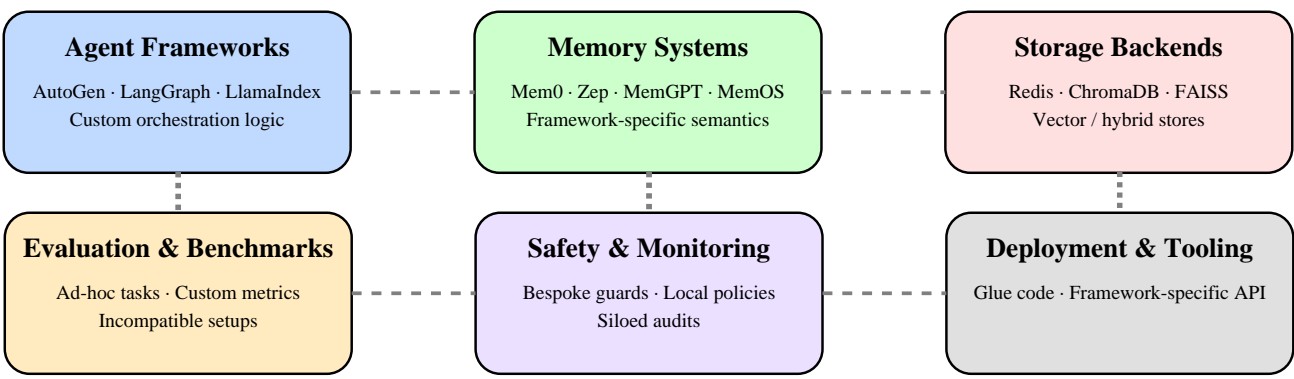

*Figure 1.* **The fragmented landscape of agentic memory systems.** Contemporary systems exhibit divergent implementations across agent frameworks, memory architectures, storage backends, evaluation protocols, safety mechanisms, and deployment tools. Incompatible or bespoke interfaces result in duplicated effort, integration overhead, and siloed safety practices. This motivates our paper's position.

infrastructure where prolonged fragmentation delayed consolidation, slowed progress, and increased systemic risk (Gu & Li, 2022; JC-Podcast, 2025).

Calls for standardisation often arise as a response to such fragmentation (Jeon, 2024). Yet rigid or premature standardisation carries its own risks in a rapidly evolving research area: it can prematurely lock in design decisions, constrain exploration, and privilege some architectural assumptions (Raza et al., 2025). Treating fragmentation and standardisation as the only two options presents a false dichotomy.

**Position. We argue that neither uncoordinated fragmentation nor rigid standardisation is a desirable path for agentic memory systems. Instead, the field should prioritise compatibility-first design, centred on shared interfaces and minimal common abstractions that enable heterogeneous memory systems to interoperate without constraining internal architectural choices.**

Compatibility at the interface level offers a third path that captures many of the benefits commonly associated with standardisation, such as interoperability, reduced redundancy and shared benchmarks and safety practices, while preserving the flexibility necessary for continued innovation. By focusing on common contracts rather than uniform implementations, compatibility-first design enables pluralism in memory architectures without limiting progress.

The remainder of our paper analyses the current landscape of agentic memory and outlines compatibility-first design, followed by a discussion of alternative viewpoints, high-level principles, concrete next steps and our conclusions.

## 2. Background and Current Limitations

Agentic memory systems have emerged through a combination of academic research prototypes and production-oriented frameworks, resulting in a diverse but highly fragmented landscape. Systems such as Mem0 (Chhikara et al.,

2025), MemOS (Li et al., 2025), Zep (Rasmussen et al., 2025), and MemGPT (Packer et al., 2023) implement memory as a core architectural component, while broader agent frameworks such as AutoGen (Microsoft, 2023), LangGraph (LangChain, 2025), and LlamaIndex (Liu, 2022) provide extensible abstractions that include memory as one subsystem among many. Despite targeting similar functionalities, such as storage, retrieval, compression, update, and access control, these systems differ in their interfaces, internal representations, and assumptions about execution context.

Fig. 1 illustrates architectural divergence across contemporary agentic memory systems. While diversity in design is expected in a rapidly evolving research area, the absence of shared abstractions or compatibility layers has led to widespread reinvention of core components. As a result, similar mechanisms are repeatedly reimplemented across incompatible frameworks, limiting reuse and complicating integration. Here, we analyse the practical costs of fragmentation, focusing on technical debt, safety/security risks, performance, memory consistency, and evaluation challenges.

### 2.1. Technical Debt and Interoperability Barriers

A direct consequence of architectural isolation is growing technical debt associated with maintaining bespoke integrations (Sousa et al., 2023). Memory systems are often tightly coupled to specific agent frameworks or storage backends, requiring custom adapters when components are combined. For example, even for the conceptual operation of storing a memory entry, contemporary systems (Mem0, LangGraph, AutoGen, MemGPT) expose fundamentally different interfaces, forcing developers to either reimplement capabilities or maintain $N \times M$ adapter functions for $N$ frameworks and $M$ memory backends; therefore, their integration can require non-trivial glue code that must be maintained as each framework evolves (Hamadi, 2025).

Based on historical evidence in adjacent areas, these costs

*Table 1.* Recent memory poisoning attacks on agentic systems demonstrate high success and highlight the fragmentation of defences.

| Attack | Target Systems | ASR | Defense Status |
|---|---|---|---|
| MINJA | GPT-4/4o (EHR, RAP, QA) | 60–90% | Existing defenses ineffective (Llama Guard, sanitization) |
| AgentPoison | RAG knowledge bases | High | Requires direct system access (isolation-based only) |
| Indirect Injection | Amazon Bedrock agents | Demonstrated | Framework-specific patches (non-transferable) |
| Email Poisoner | Custom email agent | 40–80% | Siloed implementation (per-system basis) |

*Note:* ASR: Attack Success Rate (victim producing malicious outputs). ASR varies by agent type and dataset configuration.

scale superlinearly as the number of isolated systems grows (E-Fiscal, 2013). Developers are frequently forced to choose between rewriting existing components or maintaining fragile integration layers, both of which slow down experimentation and deployment. This dynamic mirrors earlier phases of software infrastructure development in which a lack of compatibility between systems delayed consolidation and increased maintenance burdens (Feitosa et al., 2024).

### 2.2. Safety and Security Risks

Fragmentation also complicates safety and security practices (Lazer et al., 2026). Memory systems often handle sensitive information, persistent state, and long-term context, making them a critical attack surface for agentic AI systems (Microsoft AI Red Team, 2025). However, when systems are developed and deployed in isolation, vulnerability discovery, auditing, and patching occur in silos (Wan et al., 2024). Prior analyses of agentic AI failure modes indicate systemic vulnerabilities due to delayed patch deployment, inconsistent access control, and insufficient auditing (Microsoft AI Red Team, 2025; Kurshan et al., 2025). Fragmentation exacerbates these issues by preventing shared defensive practices, such as coordinated audits, common threat models, or reusable access-control mechanisms (Chen & Lu, 2025). While closed-source development further limits external scrutiny, even open systems suffer when interfaces and assumptions are incompatible across implementations (Gordon, 2025; Hartung, 2025).

Recent research has demonstrated the severity of these vulnerabilities through concrete attacks on agentic memory systems. Tab. 1 summarises major memory poisoning attacks from 2024–2025, revealing consistently high success rates and fragmented defensive responses. Attacks such as MINJA (Dong et al., 2025) achieved attack success rates exceeding 60%, while others targeted RAG knowledge bases (Chen et al., 2024) or enabled silent data exfiltration (Unit42 Team, 2025). Critically, existing defences such as Llama Guard and memory sanitisation proved ineffective across multiple foundation models (including GPT-4o-mini, Gemini-2.0-Flash, and Llama-3.1-8B-Instruct) (Sunil et al., 2026; Microsoft AI Red Team, 2025).

As shown in Tab. 1, each affected system required independent defensive solutions with non-transferable, framework-specific patches. if these memory systems shared compatible access-control interfaces, a common defensive primitive could be developed once, audited thoroughly, and deployed across all systems. Instead, fragmentation necessitates siloed security responses, prolongs exposure windows, and increases the attack surface. Similar patterns appear in prompt injection defences, role-based access control, and audit logging: without shared interfaces, security improvements remain isolated, and vulnerabilities persist longer. Tab. 2 compares the current fragmented systems with a compatibility-first ecosystem from a security perspective.

### 2.3. Suboptimal and Inconsistent Performance

Reported performance of agentic memory systems varies widely across benchmarks and task settings, and sometimes even hard to reproduce (Chalef & Rasmussen, 2025). Some systems demonstrate strong results in narrowly defined environments, while struggling in dynamic or long-horizon tasks (Schwartz et al., 2025). For example, existing evaluations indicate that several memory systems that utilise very large language models achieve less than 30% success (e.g. F1 Score) in challenging interactive settings (like multi-hop conversation analysis), a level that is insufficient for many real-world applications (Chhikara et al., 2025).

This variability is not solely due to algorithmic limitations. Fragmented development makes it difficult to transfer optimisations, compare design trade-offs, or systematically evaluate how specific memory mechanisms contribute to performance. Hence, improvements are often local rather than cumulative across the field (Hu et al., 2025a).

### 2.4. Memory Conflicts and State Consistency

Managing concurrent access and updates remains a major challenge in multi-agent or asynchronous settings (Rezazadeh et al., 2025; Cauri, 2025). Fragmented architectures rely on ad-hoc solutions for conflict resolution, versioning, and rollback, leading to inconsistent states (Han et al., 2024). Poorly coordinated update mechanisms can significantly degrade agent behaviour (DeChant, 2025; Yuen et al., 2025). The absence of shared abstractions makes it hard to reason about correctness or reuse robust solutions across systems.

*Figure 2.* Fragmentation and rigid standardisation represent two extremes in agentic memory's design. Fragmentation maximises flexibility but incurs technical debt, safety risks, and evaluation challenges; rigid standardisation improves coordination but risks premature design-space closure. Compatible design occupies a middle ground, enabling interoperability while preserving architectural diversity.

### 2.5. Lack of Comparable Evaluation

Finally, fragmentation undermines meaningful evaluation and comparison. Memory systems are often assessed using incompatible benchmarks, metrics, or experimental protocols, making claims of superiority difficult to verify (Markovic, 2025). Even when similar benchmarks are used, differences in task formulation or evaluation criteria can lead to contradictory conclusions (Bian et al., 2026).

Public disagreements over performance claims (Chalef & Rasmussen, 2025), for instance, between systems evaluated on ostensibly comparable agent-memory benchmarks, illustrate how incompatible evaluation pipelines obscure genuine progress. Without shared evaluation conventions, it is difficult for the community to identify which design choices generalise across tasks and domains (Hu et al., 2025a).

Taken together, these limitations suggest that the current trajectory of agentic memory systems, i.e., increasing fragmentation, imposes growing costs on scalability, safety, and scientific progress (Mireshghallah et al., 2025). Next, we argue that these challenges are best addressed not via rigid standardisation, but with compatible design centred on shared interfaces and minimal common abstractions.

## 3. Why Compatibility Over Fragmentation or Standardisation

The limitations discussed in Sec. 2 arise not from insufficient innovation in agentic memory systems, but from a lack of coordination at the level of interfaces and abstractions. While calls for standardisation often emerge in response to fragmentation (Miebach, 2024; Channing & Ghosh, 2025), rigid or premature standards risk constraining exploration in a rapidly evolving research area. In this section, we argue that compatibility-first design offers a principled alternative: enabling interoperability, reuse, and coordinated safety practices without imposing uniform internal architectures.

### 3.1. Enabling Modularity Without Uniformity

Compatibility-first focuses on shared interface contracts rather than shared implementations. By defining minimal, stable abstractions for core memory operations like storage, retrieval, versioning, and access control, systems can interoperate while retaining freedom to optimise or experiment internally (Kang et al., 2025; MemMachine Team, 2025).

This approach enables modular composition of memory components across frameworks. Rather than reimplementing retrieval or update mechanisms, developers can reuse compatible modules or swap components as requirements change. Importantly, such modularity does not require agreement on a single memory architecture; it requires only that systems speak a common language at their boundaries. As demonstrated in other areas of machine learning infrastructure, interface compatibility facilitates cumulative progress by allowing improvements in one system to benefit others (Balch et al., 2023; Huang et al., 2025).

### 3.2. Reducing Technical Debt and Integration Costs

Many of the integration costs identified in Sec. 2.1 stem from incompatible assumptions about memory representation and lifecycle management. Compatibility-first design reduces these costs by shifting integration efforts from bespoke glue code towards reusable adapters with shared contracts.

Over time, this transition reduces the marginal cost of adopting new memory components or evaluation tools, making experimentation cheaper and more scalable. Rather than consolidating the ecosystem around a single framework, compatibility encourages convergence at the level of interfaces, allowing multiple systems to coexist while remaining composable. This dynamic has historically enabled both rapid innovation and widespread adoption in other software ecosystems and has become a priority in the age of AI agents (Yang et al., 2025; Chan et al., 2025).

### 3.3. Evidence from Adjacent Domains

Interface-level compatibility has proven effective in addressing fragmentation in analogous software ecosystems:

**ML Model Interchange (ONNX) (Facebook & Microsoft, 2017).** The fragmentation of deep learning frameworks, such as PyTorch (Paszke et al., 2019), TensorFlow (Abadi et al., 2016), JAX (Frostig et al., 2019) and MXNet (Chen et al., 2015), created similar integration challenges. ONNX (Open Neural Network Exchange) addressed this through a shared computational graph format and minimal operator set, not by standardizing training APIs or internal tensor representations. Models trained in PyTorch can now be deployed in TensorFlow Serving (Olston et al., 2017), TensorRT (NVIDIA Team, 2016), or custom runtimes, enabling cumulative tooling (optimizers, quantizers, profilers) without dictating how frameworks implement backpropagation or automatic differentiation.

**Container Interfaces (OCI) (Open Containers Team, 2015).** Before the Open Container Initiative (OCI), container runtimes, such as Docker (Merkel, 2014), rkt (Polvi, 2014), LXC (Graber, 2014), were fragmented with incompatible formats. OCI defined minimal interface specifications enabling interoperability across Docker, containerd (Docker Team, 2015), CRI-O (Red Hat Team, 2016), and others while maintaining distinct internal architectures. We explain this further in Sec. B.

**ML Libraries (HuggingFace's Transformers (Wolf et al., 2019)).** Transformers library demonstrates interface compatibility at a smaller scale, which we discuss more in Sec. B.

**Key Pattern.** In each case, compatibility at narrow interface boundaries unlocked composability and cumulative progress without requiring architectural convergence. The same pattern applies naturally to agentic memory systems: shared contracts for storage, retrieval, and update enable ecosystem benefits while preserving internal diversity.

### 3.4. Coordinated Safety & Security Practices

Compatibility also plays an important role in improving safety and security (NCSC & CISA, 2023). Fragmented systems hinder shared threat modelling, reusable access-control mechanisms, and coordinated auditing (Marshall et al., 2019; Von der Assen et al., 2024). In contrast, compatible interfaces allow security-relevant components, such as permission models, logging mechanisms, or conflict-resolution strategies, to be reused and independently evaluated across systems (Schneider et al., 2024).

Crucially, compatibility does not require that all systems adopt identical security policies or implementations. Instead, it enables the development of shared defensive primitives and auditing tools that can be applied consistently, even

as internal architectures diverge (Zdun & Avgeriou, 2008; Huffmire et al., 2010). This supports collective defence without requiring centralised control or uniform design.

### 3.5. Improving Comparability & Performance Gains

Performance limitations in agentic memory systems are exacerbated by fragmented evaluation pipelines (Wei et al., 2025b; Bian et al., 2026). When systems expose compatible interfaces, benchmarks and testing harnesses can be reused across frameworks, enabling more meaningful comparison and cumulative optimisation (Schwartz et al., 2025).

Compatibility-first design allows researchers to isolate and evaluate the contribution of specific memory mechanisms, facilitating the transfer of successful ideas across implementations. Over time, this supports a shift from framework-specific optimisation toward field-level progress, where improvements generalise beyond a single system.

### 3.6. Economic and Ethical Implications

Compatibility-first design's economic and ethical implications are immense. By reducing redundant effort, **compatible ecosystems lower barriers to entry for smaller teams and academic groups**, reducing dependence on vertically integrated platforms and preventing concentration of influence over safety-critical infrastructure. **This enables safety improvements**, evaluation tools, and defensive mechanisms to propagate across systems, promoting coordination without coercion while preserving research diversity.

### 3.7. Timeliness

We think that the present moment represents a critical window for compatibility-first design. First, **agentic memory systems have reached minimal required structural maturity**. Core functional roles (storage, retrieval, update, versioning, access control) have stabilised, while internal implementations continue to diverge, creating ideal conditions for interface coordination without freezing research. Second, **fragmentation costs are now measurable** (see Sec. 6.4) and scaling superlinearly: integration overhead, duplicated effort, delayed security patches, and irreproducible evaluations impose concrete burdens that can be arrested before calcification occurs. Third, **the ecosystem is ready to absorb compatibility**; researchers increasingly demand composability and cross-system comparison, and lightweight adapters and shared interfaces deliver immediate local benefits: reduced integration effort, shared tooling, and easier evaluation.

Crucially, key research questions about memory representations, update semantics, and safety guarantees remain unsettled, making rigid standardisation premature. The field has sufficient shared understanding to define minimal inter-

operability contracts, yet remains early enough that rigid standards would freeze active research. Compatibility-first design aligns systems for interoperability while keeping flexibility for research diversity. Delaying could lead to stagnation, while standardising too early risks premature closure. Acting now strikes the necessary balance between fragmentation and rigid standardisation (See Fig. 2).

Taken together, these arguments suggest that compatibility-first design directly addresses the fragmentation-related limitations identified in Sec. 2, while avoiding the risks associated with rigid or premature standardisation. In the next section, we examine alternative viewpoints, including arguments favouring continued fragmentation or early standardisation, and discuss their respective limitations.

## 4. Alternative Views and Why They Fall Short

The argument for compatibility-first design is best understood in contrast to several credible alternative viewpoints that currently shape the development of agentic memory systems. Below, we examine three such perspectives:

### 4.1. Embracing Fragmentation to Maximise Flexibility

One common view is that fragmentation is a natural and even desirable feature of a rapidly evolving research area. According to this perspective, independent development maximises flexibility, encourages architectural diversity, and avoids premature convergence on suboptimal abstractions. From this point of view, attempts at coordination risk slowing down innovation or privileging dominant players.

Although this argument correctly highlights the value of exploration, it underestimates the cumulative costs of uncoordinated fragmentation. As discussed in Sec. 2, the absence of shared interfaces leads to repeated reinvention of core mechanisms, increased integration costs, and limited reuse of safety and evaluation tools. Over time, these inefficiencies reduce the pace of progress rather than accelerating it. Compatibility-first design preserves architectural freedom while mitigating these costs by enabling systems to interoperate without enforcing a uniform internal structure.

### 4.2. Early or Rigid Standardisation

At the opposite extreme, some argue that the challenges posed by fragmentation can only be addressed through early standardisation. A single unified standard, they contend, would simplify integration, improve security practices, and enable fair comparison across systems.

However, in a field where design choices around memory representation, update semantics, and agent interaction remain active research questions, rigid standardisation risks locking in assumptions that may later prove limiting. Historical precedents in computing (loss of OSI standardisation to more flexible TCP/IP suite (Russell, 2006) or CODA-SYL's network database model (Haigh, 2006), which was displaced by more flexible relational model) suggest that premature standards can slow innovation, entrench early design decisions, and discourage experimentation. Moreover, enforcing a single standard can centralise power and reduce the diversity of approaches explored by the community.

Compatibility-first design offers a more flexible alternative: coordination at the interface level rather than at the full system architecture level (see Sec. C for detailed clarifications). By focusing on shared contracts instead of uniform implementations, the field can gain many of the benefits of standardisation: interoperability, comparability, and reuse, without prematurely closing the design space.

### 4.3. Vertically Integrated Proprietary Systems

A third viewpoint prioritises vertically integrated, often proprietary systems, arguing that tight control enables faster commercialisation, more coherent design, and improved reliability. From this perspective, fragmentation and open coordination introduce complexity that slows deployment and weakens accountability. While such systems can be effective within bounded environments, they limit composability and external scrutiny. Proprietary integration constrains interoperability with other tools and frameworks, making it difficult for improvements in safety, evaluation, or memory management to propagate beyond organisational boundaries. Furthermore, reliance on closed ecosystems can exacerbate concentration of influence over safety-critical infrastructure.

Compatibility-first design does not preclude proprietary implementations, but limits their isolation. By encouraging shared interfaces, even closed systems can participate in broader ecosystems of evaluation tools, benchmarks, and defensive mechanisms, balancing commercial incentives with collective progress.

### 4.4. Summary

Each of these alternative views captures an important concern: flexibility, coordination, or speed. However, when pursued in isolation, they introduce significant trade-offs that hinder scalability, safety, or cumulative progress. Compatibility-first design addresses these tensions by enabling coordination without coercion, supporting interoperability and reuse while preserving freedom to innovate. This positioning motivates the design principles and adoption strategies outlined in the next section. We provide a detailed clarification with similar analogies in Sec. C.

## 5. Blueprint for Compatibility-First Memory

Compatibility-first design requires more than high-level principles; it requires concrete interface definitions, coor-

dination mechanisms, and adoption pathways that enable interoperability without constraining internal system design. In this section, we outline a high-level blueprint that operationalises compatibility-first agentic memory systems while preserving architectural pluralism.

### 5.1. Minimal Interface Abstractions

Minimal, stable interface abstractions for common memory operations lie at the core of our design. These abstractions should specify what a memory component exposes rather than how it is implemented. Core examples include abstract interfaces for storage and retrieval (writing, querying, and deleting memory entries, independent of backend choice), optional interfaces for update and versioning (snapshotting, rollback, and conflict resolution), and standardised hooks for access control (enforcing permissions and roles in multi-agent settings without prescribing specific security models). Systems that implement these interfaces can interoperate, regardless of internal design choices.

**Concrete Example: Memory Interface and Adapter** To illustrate our compatibility-first design, Algorithm 1 demonstrates how one of the popular existing systems called `Mem0` (Chhikara et al., 2025) can be adapted with minimal code.

---

**Algorithm 1** Adapting `Mem0` to MemoryInterface

```
class Mem0Adapter(MemoryInterface):
  """Mem0 extending MemoryInterface."""
  def __init__(self, config: dict):
    from mem0 import Memory
    self.mem0 = Memory.from_config(config)
  def store(self, content: str, metadata: dict,
            agent_id: str) -> str:
    result = self.mem0.add(
      messages=[{"role": "user", "content": content}],
      user_id=agent_id, metadata=metadata)
    return result['id']
  def retrieve(self, query: str, agent_id: str,
               limit: int = 5):
    results = self.mem0.search(
      query=query, user_id=agent_id, limit=limit)
    return [MemoryEntry(id=r['id'],content=r['memory'],
      metadata=r.get('metadata', {})) for r in results]
  def update(self, entry_id: str, content: str,
             metadata: dict) -> bool:
    return self.mem0.update(memory_id=entry_id,
      data=content)
  def delete(self, entry_id: str) -> bool:
    self.mem0.delete(memory_id=entry_id)
    return True
```

---

This adapter enables `Mem0` to interoperate with any tool expecting the `MemoryInterface` contract without modifying `Mem0`'s internals. Approximately 20-30 lines of wrapper code provide full compatibility. The key insight: **compatibility emerges through thin translation layers, not architectural rewrites**.

### 5.2. Reference Implementations and Adapters

Interface specifications should be accompanied by lightweight reference implementations that serve as exe-

cutable documentation. Adapters wrap existing systems to conform to shared interfaces, allowing compatibility to emerge incrementally without costly rewrites, enabling existing frameworks to participate in broader interoperable ecosystems with minimal disruption.

### 5.3. Shared Evaluation and Benchmarking Hooks

Compatibility-first design also enables shared evaluation infrastructure. When systems expose compatible interfaces, benchmarking harnesses and evaluation tools can be reused across frameworks. Rather than defining a single benchmark suite, compatibility facilitates a family of interoperable evaluation pipelines that measure memory performance, robustness, and safety under comparable conditions.

Such hooks make it possible to evaluate specific design choices, such as update strategies or retrieval mechanisms, independently of the surrounding framework. Over time, this supports cumulative performance improvements and clearer attribution of empirical gains.

### 5.4. Lightweight Coordination and Governance

Maintaining compatibility does not require heavy-handed governance. Instead, lightweight coordination mechanisms can help align interface evolution while preserving flexibility, including open proposals for interface extensions or revisions, small rotating working groups focused on interoperability, security, or evaluation, and versioned interface specifications with clear deprecation paths. Importantly, participation in such coordination efforts should remain voluntary. Influence emerges through adoption rather than enforcement, allowing successful interfaces to become de facto standards without formal mandates.

### 5.5. Adoption Pathways

Compatibility-first systems are most likely to gain traction when they offer immediate, local benefits. Early adoption can be incentivised through drop-in adapters that enable existing systems to interoperate with minimal changes, shared benchmarks that reduce evaluation effort, and reusable safety and monitoring components built atop common interfaces. By focusing on incremental gains rather than wholesale ecosystem redesign, compatibility-first design encourages gradual convergence while respecting existing investments and research trajectories.

## 6. A Compatibility-First Call to Action

### 6.1. Define and Test Shared Interfaces

Researchers are well positioned to initiate compatibility-first progress by focusing on interfaces rather than full systems. We encourage publishing minimal interface spec-

ifications alongside new memory systems, clarifying assumptions about storage, retrieval, update, and access control. Reference implementations and adapters that allow existing frameworks to interoperate without rewrites can accelerate adoption. Experiments and benchmarks should explicitly evaluate memory components through shared interfaces, enabling meaningful comparison across systems. Workshops, open-source repositories, and living documents hosted alongside preprints serve as low-friction venues for iterating on these interfaces. Even partial adoption—such as exposing a single compatible retrieval API—can meaningfully reduce fragmentation.

### 6.2. Reward Interoperability in Deployment

Practitioners deploying agentic systems shape ecosystem incentives through their architectural choices. Preferring memory components and frameworks that expose well-documented, interoperable interfaces sends market signals that compatibility matters. Contributing adapters, integration examples, and failure reports surfaces real-world compatibility issues that guide interface refinement. Sharing deployment lessons—particularly around safety, performance, and maintenance costs attributable to incompatible memory systems—helps the community understand fragmentation's true costs. By favoring interoperable components in practice, practitioners help compatibility emerge as a competitive advantage rather than an external constraint.

### 6.3. Incentivise Compatibility

Funders and evaluation bodies can accelerate compatibility-first adoption through modest incentive adjustments. Supporting projects that prioritise reusable interfaces, adapters, and shared benchmarks alongside novel memory designs signals that interoperability is valued. Encouraging proposals to articulate how their systems interoperate with existing tools or contribute to ecosystem-level compatibility makes coordination explicit. Funding maintenance, auditing, and benchmarking efforts that benefit multiple memory systems—rather than single frameworks—recognizes infrastructure work as a legitimate research contribution. Such incentives need not mandate standards; they simply acknowledge that interoperability and reuse are valuable research outputs in their own right.

### 6.4. Measuring Progress

Compatibility-first design succeeds to the extent it achieves measurable reductions in fragmentation costs and enables cumulative ecosystem growth. We propose tracking progress across four dimensions:

**Adoption metrics:** measure the uptake of shared interfaces (percentage of new memory systems exposing compatible

APIs, availability of cross-framework adapters, growth in multi-framework projects).

**Technical debt reduction:** tracks integration efficiency (reduction in integration code as a percentage of the codebase, decrease in memory-related bugs, reduction in time to fix upstream API changes).

**Ecosystem health:** assesses collective capabilities (number of shared evaluation benchmarks compatible with multiple systems, growth in reusable safety and monitoring components, citations of interface specifications in new research).

**Community coordination:** gauges collaborative activity (participation in interface design forums, contributions to reference implementations and adapters, engagement with specification documents).

These metrics are intentionally lightweight and observable through public repositories and publication records. Communities adopting compatibility-first principles can define specific targets appropriate to their context and maturity. Compatibility-first ecosystems rarely emerge through top-down design; instead, they converge as interfaces that solve real problems gain adoption. By aligning research practices, deployment incentives, and funding criteria around compatibility, the community can reduce fragmentation while preserving the diversity and experimentation that drive progress. Regular retrospectives at major conferences can assess progress and refine interface designs based on empirical adoption patterns.

## 7. Conclusions

Agentic memory systems are becoming foundational to capable AI agents, yet their development is constrained by a false dichotomy. Uncoordinated fragmentation leads to redundant effort, rising technical debt, inconsistent safety practices, and limited comparability. Rigid standardisation risks constraining experimentation and locking in design assumptions in a rapidly evolving field.

In this paper, we have argued for compatibility-first design as a principled alternative. By focusing on shared interfaces and minimal common abstractions rather than uniform implementations, this approach enables interoperability, reuse, and coordinated safety practices without foreclosing architectural diversity. Crucially, this progress can emerge incrementally through adoption rather than top-down mandates. The future of agentic memory systems will be determined not by fragmentation or standardisation alone, but by the field's ability to coordinate without constraining innovation. Compatibility-first design offers a practical foundation for achieving this balance, fostering collaboration and preserving flexibility to drive the field toward more robust and secure, yet innovative AI systems.

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

# Appendix

## A. Fragmentation vs Compatible Design: A Security Perspective

*Table 2.* Fragmentation amplifies security vulnerabilities: incompatible systems require independent defensive solutions, leading to delayed and inconsistent protection.

| Security Dimension | Fragmented Systems | Compatible Systems |
|---|---|---|
| Vulnerability Discovery | Siloed testing; inconsistent coverage | Shared security tools; comprehensive coverage |
| Patch Development | Independent solutions per framework | Unified defensive primitives |
| Deployment Timeline | Delayed (weeks to months); requires per-system adaptation | Rapid (days); adapter-level updates |
| Defense Transfer | Non-transferable; manual reimplementation | Reusable across compatible interfaces |
| Audit Coordination | Isolated audits; no shared threat models | Coordinated audits; common security primitives |
| Attack Surface | Multiplicative across $N$ systems | Contained at interface boundaries |

## B. Further Evidence of Compatibility Success from Adjacent Domains

### B.1. Container Interfaces (OCI)

Before the Open Container Initiative (OCI), container runtimes were fragmented; Docker (Merkel, 2014), rkt (Polvi, 2014), LXC (Graber, 2014) used incompatible formats. Rather than mandating a single implementation, OCI defined minimal interface specifications for container images and runtimes. This enabled Docker, containerd (Docker Team, 2015), CRI-O (Red Hat Team, 2016), and others to interoperate while maintaining distinct internal architectures. Today, Kubernetes (Burns et al., 2016) can run containers from any OCI-compliant runtime, and image formats transfer seamlessly across registries, all without constraining runtime innovation.

### B.2. HuggingFace's Transformers

HuggingFace Transformers demonstrates interface compatibility at a smaller scale. By defining common interfaces for tokenisers, models, and trainers, the library enables over 100,000 models to work with shared evaluation tools, training scripts, and deployment pipelines, despite vast diversity in architectures (transformers (Vaswani et al., 2017), diffusion models (Ho et al., 2020), vision encoders (Dosovitskiy et al., 2021)). New model classes require only implementing the base interface to inherit the full ecosystem.

## C. What is Compatibility-First Design and How Does It Differ from Existing Frameworks?

A common question is: *"Don't frameworks like AutoGen, LangGraph, and LlamaIndex already provide memory abstractions? What is compatibility-first design offering that's different?"* This appendix clarifies the distinction and addresses what lies under the scope of our proposed compatibility-first design and what does not.

### C.1. The Key Distinction: Ecosystem Philosophy vs. Framework Implementation

**Compatibility-first design is not a competing framework.** It is a design philosophy and coordination strategy for the entire ecosystem of agentic memory systems. The crucial differences are:

- **Existing frameworks** (AutoGen, LangGraph, LlamaIndex, Mem0, Zep) provide *internal* memory abstractions within their own systems. Each framework has its own API, data models, and assumptions about memory lifecycle.

- **Compatibility-first design** advocates for *shared interface contracts across* frameworks, enabling heterogeneous

systems to interoperate without requiring users to commit to a single framework.

## C.2. Compatibility-First Contract: Scope and Non-Goals

Compatibility-first design coordinates agentic memory systems at the level of interfaces rather than implementations. To avoid ambiguity, we explicitly delineate which aspects of memory systems fall within the scope of interface-level compatibility and which aspects are intentionally left unconstrained.

*Table 3.* Interface-level elements coordinated under compatibility-first design and aspects that remain explicitly out of scope.

| Category | Coordinated at the Interface Level | Explicitly Out of Scope |
|---|---|---|
| Core memory operations | `store`, `retrieve`, `update`, `delete` method interfaces | Internal storage layouts or data structures |
| Inputs and outputs | Request–response shapes including content, metadata, and identifiers | Memory representations (e.g., text, embeddings, graphs, key–value stores) |
| Identity and scope | Agent and user scoping semantics | Cross-agent sharing or visibility policies |
| Versioning | Optional hooks for snapshotting, rollback, and conflict detection | Conflict-resolution strategies and consistency models |
| Access control | Policy-agnostic permission and authorization hooks | Concrete security policies or threat models |
| Error handling | Standardized failure modes and error signals | Recovery mechanisms or retry strategies |
| Evaluation hooks | Compatibility with shared benchmarking harnesses | Task definitions, datasets, or scoring metrics |

Compatibility-first design deliberately avoids constraining architectural or algorithmic choices that remain active research questions. In particular, it does not mandate memory schemas, retrieval algorithms, compression or reflection mechanisms, learning rules, safety policy semantics, or execution models. Systems are free to diverge internally, provided they satisfy the shared interface contracts. This design agenda enforces the interface compatibility only where fragmentation has demonstrable cost, while the surrounding design space remains open to experimentation.

## C.3. Concrete Example: What Is Missing Today

Consider a researcher who wants to:

1. Store memories using Mem0's hierarchical memory system

2. Retrieve them in an AutoGen multi-agent workflow

3. Evaluate performance using a LangGraph-based benchmark

4. Apply security primitives developed for LlamaIndex

**Current reality:** This requires writing substantial glue code, handling incompatible data formats, and reimplementing logic at each boundary. Each framework's memory system speaks a different language.

**With compatibility-first design:** If all frameworks exposed a shared `MemoryInterface` (as illustrated in Algorithm 1), the same memory backend could be swapped seamlessly across frameworks via lightweight adapters, benchmarks could run against any compatible system, and security tools would work universally.

## C.4. Why Existing Framework Abstractions Are Insufficient

Individual frameworks do provide memory abstractions, but these abstractions serve different goals:

**Framework-Internal Abstractions** AutoGen's memory API is designed for AutoGen's agent orchestration model. LlamaIndex's memory system is optimized for its query engine architecture. These are *vertical integration* solutions that work excellently within their ecosystems but do not facilitate *horizontal interoperability* across ecosystems.

**Compatibility-First Abstractions** are specifically designed to:

- Be framework-agnostic (no assumptions about agent orchestration, execution model, or surrounding infrastructure)

- Enable composability (mix-and-match memory backends, retrieval strategies, and evaluation tools from different sources)

- Support incremental adoption (existing systems can expose compatible interfaces via thin adapters without architectural rewrites)

- Facilitate ecosystem-wide benefits (shared benchmarks, reusable security primitives, coordinated auditing)

## C.5. The Analogy: Web Browsers vs. Web Standards

A useful analogy:

- **Existing frameworks** are like different web browsers (Chrome, Firefox, Safari). Each has excellent internal architecture and features, but they compete as complete products.

- **Compatibility-first design** is like advocating for web standards (HTML, CSS, HTTP, DOM APIs). Standards do not replace browsers; they enable all browsers to render the same websites, allow developers to build applications once, and permit innovation in browser internals while maintaining interoperability.

Similarly, compatibility-first design does not ask researchers to abandon AutoGen or LlamaIndex. It asks these frameworks to expose shared interface contracts so that memory components, benchmarks, safety tools, and research innovations can flow freely across the ecosystem.

## C.6. What Compatibility-First Design Requires

Operationalizing this vision requires:

1. **Minimal interface specifications**: Community-developed contracts for core memory operations (store, retrieve, update, delete, version control, access control). These should be minimal, stable, and framework-agnostic.

2. **Reference implementations & adapters**: Lightweight wrappers (like Algorithm 1's `Mem0Adapter`) that expose existing frameworks' memory systems through the shared interface. Crucially, these adapters require ∼20-30 lines of code, not architectural rewrites.

3. **Interoperable evaluation infrastructure**: Benchmarks and testing harnesses that accept any memory system implementing the shared interface, enabling fair cross-framework comparison.

4. **Shared safety primitives**: Reusable access control, auditing, and defensive components built atop common interfaces, preventing the siloed security practices.

5. **Voluntary coordination mechanisms**: Lightweight governance (e.g., working groups, versioned specs, open RFCs) that aligns interface evolution without mandating participation or constraining internal innovation.

## C.7. Why This Matters: Fragmentation Costs Are Real and Growing

As documented in Section 2, current fragmentation imposes measurable costs:

- **Technical debt**: Maintaining $N \times M$ adapters for $N$ frameworks and $M$ backends (Section 2.1)

- **Security vulnerabilities**: Siloed auditing and non-transferable patches (Table 1, Section 2.2)

- **Irreproducible evaluations**: Incompatible benchmarks obscure genuine progress (Section 2.5)

- **Wasted effort**: Reimplementing similar retrieval, compression, and update logic across isolated systems

Individual frameworks cannot solve these ecosystem-level problems through internal abstractions alone. Coordination at the interface level is necessary.

**C.8. Precedent: Where Interface Compatibility Has Succeeded**

This approach is not novel; it reflects successful patterns from adjacent domains:

- **ONNX** (Section 3.3): Models trained in PyTorch deploy in TensorFlow Serving or TensorRT via a shared computational graph format, without standardizing training APIs or internal tensor representations.

- **Open Container Initiative (OCI)** (Appendix B.1): Docker, containerd, CRI-O interoperate despite divergent internal architectures because they expose compatible image and runtime interfaces.

- **HuggingFace Transformers** (Appendix B.2): Over 100,000 models with vastly different architectures share evaluation tools, training scripts, and deployment pipelines via common tokenizer/model/trainer interfaces.

In each case, interface compatibility unlocked composability and cumulative progress *without* requiring architectural convergence or displacing existing implementations.

**C.9. Summary: Complementary, Not Competitive**

Compatibility-first design is **complementary** to existing frameworks, not competitive with them. It does not ask users to choose "our framework" over AutoGen or LlamaIndex. Instead, it asks the community—including framework developers, researchers, and practitioners—to coordinate on minimal shared interfaces that preserve each framework's strengths while enabling ecosystem-wide benefits: interoperability, reusable tooling, fair evaluation, and coordinated safety practices.

The goal is a future where researchers can:

- Use AutoGen's excellent orchestration with Mem0's memory backend

- Evaluate any memory system against shared benchmarks

- Apply vetted security primitives universally

- Transfer innovations across frameworks without reimplementation

This requires coordination, not consolidation. Compatibility-first design provides the conceptual and technical foundation for that coordination.

