# OpenReview forum: "Position: Compatibility-First Design Is Critical for Progress in Agentic Memory"
_ICML.cc/2026/Position_Paper_Track — Submitted to ICML 2026 Position Paper Track_

### Official Review · Reviewer_1AwF · 2026-03-13

**Significance:** 2
**Argument Clarity:** 1
**Rating:** 2
**Confidence:** 3

**Questions:**

Question:
1. What is the definition of compatibility for agent memory?
2. Could you further discuss the trade-off between effectiveness and efficiency?

**Alternative Views Section:**

No

**Compliance With Llm Reviewing Policy A Conservative:**

Affirmed.

**Discussion Potential:**

1

**Paper Summary:**

This paper considers that compatibility is the most important principle for the design of agent memory. It provides discussions from several aspects.

**Position:**

Yes

**Position In Title:**

Yes

**Related Work:**

2

**Strengths And Weaknesses:**

Pros:
The topic of this position is important for agent memory.

Cons:
1. I think the author should use the academic terms in the field agent memory. The current presentation makes it hard for readers to understand.
2. The form of citation is bad. Do NOT use this citation form (e.g., line 35-44) without enough explanation.
3. I think a clear definition of the compatibility for agent memory is required.
4. Lack the alternative views section.

**Support:**

2

---

> ### Author Rebuttal · Authors · 2026-03-31
>
> We thank reviewer 1AwF for their valuable feedback and address their points below.
>
> ### W1: Academic Terms
> We would like to clarify that while we have tried to use academic terms on agentic memory systems, our work focuses on the fragmentation/compatibility for memory systems (e.g. safety auditing, siloed evaluation, etc.); thus, the terminology is more consistent with academic literature on that area. We are happy to review the terminology if the reviewer has specific suggestions.
>
> ### W2: Citation Form
> We acknowledge the concern. Two passages list systems and benchmarks in rapid succession without contextual description, which we agree can feel opaque. We are replacing them with structured narratives to clarify each citation's functional role:
>
> Replace lines 35-47 (left column) with:
> > The current agentic memory landscape is diverse and growing rapidly. Representative systems already span several distinct roles in the ecosystem: dedicated memory architectures — such as MemGPT \citep{MemGPT}, Mem0 \citep{Mem0}, MemOS \citep{MemOS}, Mirix \citep{Mirix}, and Zep \citep{Zep} — implement memory as a first-class architectural component with explicit storage, retrieval, and consolidation mechanisms; broader agent frameworks with integrated memory modules — such as LangGraph \citep{LangGraph}, AutoGen \citep{AutoGen}, LlamaIndex \citep{LlamaIndex}, and LangMem \citep{LangMem} — treat memory as one subsystem among many within a general-purpose orchestration layer; and storage and infrastructure backends — such as Weaviate \citep{Weaviate}, Pinecone \citep{Pinecone}, and Chroma \citep{Chroma} — provide the underlying retrieval and persistence substrate on which higher-level memory systems build. This functional diversity is a sign of a healthy, innovative ecosystem. However, it also means that semantically equivalent operations — storing a fact, retrieving a relevant context, or deleting a stale entry — are exposed through mutually incompatible abstractions across systems, creating the interoperability challenges this paper addresses.
>
> We also replace lines 17-26 similarly to address their point.
>
> ### W3&Q1: Compatibility for Agentic Memory Field
> We currently address the details of the concept of compatibility-first design from multiple perspectives. Please check Section 1 (lines 90-97), 3.1 (lines 180-196), 3.4 (lines 260-273) and Appendix C (lines 920-1076). Section 5 also details design/implementation steps for achieving this compatibility.
>
> We are adding this definition to the beginning of section 3 to succinctly define compatibility:
>
> > **Compatibility for agentic memory** is a design approach that ensures interoperability across diverse memory systems **through shared interfaces and minimal common abstractions**, while preserving architectural flexibility that allows integration of novel innovations in the field. It addresses fragmentation and premature standardisation by standardising core operations (e.g., store, retrieve, and delete) via interfaces such as `MemoryInterface`, providing adapters to bridge existing systems, and facilitating shared benchmarks and safety practices. It enables (and encourages) flexibility by allowing hooks and extra specific functionalities that could be integrated (e.g. via cyclic reviews done by a working group) to allow for innovation and to prevent rigid standardisation.
>
> ### W4: Alternative Views Section
> Please refer to Section 4 (**Alternative Views and Why They Fall Short**, page 6, line 290), where we present three alternative views and argue that compatibility-first design offers the best trade-off between fostering innovation and reducing fragmentation costs.
>
>
> ### Q2: Effectiveness/Efficiency Trade-off
> We acknowledge the need to clarify this trade-off. A compatibility-first agenda provides a desirable balance between the two. Below, we provide a Gain/Cost analysis:
>
> **Interoperability**: Standardised interfaces (e.g., `MemoryInterface`) allow frameworks such as LangGraph and AutoGen to exchange data seamlessly, reducing redundant implementations and safety gaps. --> *+ Efficiency & Efficacy*
>
> **Collaboration**: Shared benchmarks and evaluation tools foster innovation and reduce technical debt. --> *+ Efficiency & Efficacy*
>
> **Adapter Overhead**: Bridging systems via adapters may introduce marginal latency or complexity (e.g., translating between memory schemas). This increase is negligible as it only adds a unifying layer/API. --> *- Efficiency* (if not managed carefully)
>
> **Design Constraints**: Minimal abstractions preserve flexibility but require careful implementation to avoid limiting novel architectures. --> *- Efficacy* (if not governed properly)
>
> **Trade-off Resolution**
>
> Compatibility-first design prioritises long-term effectiveness (avoiding duplication and safety risks) while mitigating efficiency costs through lightweight interfaces and incremental adoption. This mirrors successes such as ONNX, balancing portability with minimal performance impact.

---

> > ### Author Rebuttal · Reviewer_1AwF · 2026-04-01
> >
> > Thanks for the rebuttal, and I prefer to maintain my score.

---

### Official Review · Reviewer_i634 · 2026-03-13

**Significance:** 2
**Argument Clarity:** 3
**Rating:** 4
**Confidence:** 4

**Questions:**

1. What empirical evidence supports the claim that compatibility-first design reduces integration cost, improves safety reuse, or enhances performance comparability in practice?

2. How do the authors prevent a widely adopted compatibility layer from gradually becoming a de facto rigid standard that constrains future innovation?

3. How much of agentic memory’s core complexity can actually be captured by the proposed minimal shared interface, beyond basic storage and retrieval operations?

**Alternative Views Section:**

Yes

**Compliance With Llm Reviewing Policy A Conservative:**

Affirmed.

**Discussion Potential:**

3

**Final Justification:**

The authors clarified my main reservations and outlined a pragmatic and actionable roadmap for the agentic memory ecosystem. I keep my positive rating unchanged.

**Paper Summary:**

The authors claims that the agentic memory need improve its compatibility, instead of chasing uncoordinated fragmentation or rigid standardisation. The compatibility-first design offers a better path, where systems can share minimal interface contracts while preserving freedom in their internal implementations. Because current fragmentation causes rising technical debt, security and auditing gaps, incompatible evaluation pipelines, and repeated reinvention across frameworks. At the same time, they reject early standardisation because key questions, e.g., memory representation, update semantics, and safety guarantees, remain unresolved. The paper’s main contribution is to reframe the problem as one of interoperability, defend compatibility as a middle ground between flexibility and coordination, and propose a practical roadmap built on minimal interfaces, lightweight adapters, interoperable benchmarks, shared safety primitives, and voluntary governance.

**Position:**

Yes

**Position In Title:**

Yes

**Related Work:**

4

**Strengths And Weaknesses:**

Strengths

S1. The paper identifies a timely and practically crucial ecosystem-level problem in agentic memory, highlighting systemic challenges such as technical debt, poor interoperability, limited reuse of safety tools, and non-comparable evaluation.

S2. The authors propose a practical compatibility-first framework that is both actionable and broadly impactful, bridging research needs such as comparable evaluation with ecosystem needs.

S3. This paper involves sufficient work on agentic memory, making its analysis comprehensive and conclusive.

Weaknesses

W1. The paper primarily advances a fairly traditional software compatibility argument at the ecosystem level, but does not make a substantive direct contribution to agentic memory itself in terms of methods, mechanisms, or empirical understanding.

W2. This paper does not directly analyze potential runtime overheads or impact on utilizing agentic memory introduced by a unified interface.

W3. The proposed interface may be too shallow to capture the core complexity of agentic memory.

**Support:**

3

---

> ### Author Rebuttal · Authors · 2026-03-31
>
> We thank the reviewer for their thoughtful feedback and questions. We appreciate their interest in our paper as a timely, crucial and actionable position. Below we address their questions and concerns.
>
> ### W1: Contribution to Agentic Memory
> Our contribution lies in adapting compatibility principles to the agentic memory, where fragmentation risks stifling innovation. By framing interoperability as a prerequisite for progress (e.g., enabling shared safety audits), we provide a scaffold for future methodological advancements. This mirrors how standards such as ONNX accelerate deployment without constraining model architecture.
>
> ### W2: Runtime Overhead Analysis
> We acknowledge the need to discuss the impact of a unified interface on runtime and utilisation of agentic memory. We will add the following clarifications to our paper to address your point.
>
> **Runtime Overhead**: The adapter layer is the only overhead introduced by the unified interface. This interface/adapter may introduce marginal latency or complexity (e.g., translating between memory schemas). This increase is negligible, as it only adds a unifying layer/API. However, it is worth noting existence of a working group or a governing body is crucial for this overhead to remain negligible, as the field matures and more (complex) functionalities are added over time to prevent inefficiencies in the unifying interface.
>
> **Impact on Utilising agentic memory**: The unified interface enhances utilisation of these systems by boosting:
>
> - **Cross-framework compatibility** by allowing frameworks such as LangGraph and AutoGen to seamlessly exchange data while addressing safety/conflict issues arising from systems utilising a multi-framework solution.
> - **Comparability** of these systems by enabling smooth and fair evaluation on shared benchmarks, which enables developers to properly assess which systems to choose based on their performance criteria for specific target domains.
>
> ### W3: Interface Adequacy
>
> The proposed `MemoryInterface` intentionally focuses on core operations (store/retrieve/delete) to avoid premature constraints, which would lead to rigid standardisation and reduced innovation. The motivation for the current interface is to provide an example/blueprint and encourage discussion on a compatibility-first agenda; the community, via e.g., a working group, can adapt it over time, determine which future functionalities should be integrated as the field matures. For current advanced needs (e.g., temporal indexing or access control), we advocate for:
>
> * **Extensible Hooks**: Modular extensions (e.g., a `SafetyInterface`) to layer complexity.
> * **Community-Driven Evolution**: Interfaces expanding via proposals (e.g., GitHub RFCs) as the field matures.
>
> ### Q1: Empirical Evidence for Compatibility
> Empirical data from adjacent fields provide strong supporting evidence:
>
> * **Huggingface Transformers**: Standardised interfaces reduced NLP framework duplication as the library replaced the need for bespoke implementations of transformer models with a single API. Its impact on consolidating integration effort has been documented in [1], which shows that 80% of pretrained model descendants on HF adopt the Transformers library, indicating a reduction of duplicated effort.
>
> * **ONNX Model Interchange**:  [2] found that ONNX is the most widely adopted interoperability tool in the Deep Learning ecosystem, with developers citing reduced integration effort when moving models between frameworks as the primary motivation. Similarly, evaluating five model export formats across two ML-enabled systems, [3] confirmed that ONNX offered the most efficient integration and portability across the majority of deployment scenarios.
>
> ### Q2: Preventing Rigid Standardisation
> To prevent this, we propose:
>
> * **Voluntary Adoption**: Interfaces as opt-in tools, not mandates.
> * **Evolutionary Governance**: A community working group (similar to Python PEP or ONNX SIGs) would review interface extensions through a transparent proposal process, ensuring additions are backwards-compatible, evidence-based, and reversible.
>
>
> ### Q3: Capturing Agentic Memory Complexity
> The interface targets foundational needs that comprise the majority of use cases, since most agent-memory interactions require only core read/write operations. For advanced scenarios, frameworks can implement extensions (e.g., an `update_semantics` layer for conflict resolution, or a `SafetyInterface` for access control).
>
> [1] "[What do we know about Hugging Face?...](https://dl.acm.org/doi/fullHtml/10.1145/3674805.3686665)" ESEM 2024.
>
> [2] "[Interoperability in deep learning...](https://arxiv.org/html/2303.17708v4)" ACM SIGSOFT 2024.
>
> [3] "[How do model export formats ...](https://arxiv.org/abs/2502.00429)" IEEE/ACM CAIN 2025.

---

> > ### Author Rebuttal · Reviewer_i634 · 2026-04-03
> >
> > The authors clarified my main reservations and outlined a pragmatic and actionable roadmap for the agentic memory ecosystem. I keep my positive rating unchanged.

---

### Official Review · Reviewer_ADFN · 2026-03-14

**Significance:** 3
**Argument Clarity:** 4
**Rating:** 5
**Confidence:** 4

**Questions:**

1) I am not sure I agree that section `2.3. Suboptimal and Inconsistent Performance` and `2.5. Lack of Comparable Evaluation` are fully separate, they seem to both hinge on the lack of comprehensive evaluation for agentic memory systems. Should these be joined?
2) In this same vein, is section `2.2 Safety and Security Risks` not just another facet of this same issue. I think it might improve the clarity of the argument to group these different limitation types under "Lack of Evaluation Standards" or something like this.
3) I am skeptical of whether `2.4. Memory Conflicts and State Consistency` is actually a limitation brought about by fragmentation or that merely exists in agentic memory systems. Could you elaborate on why memory conflicts are listed in this section?

**Alternative Views Section:**

Yes

**Compliance With Llm Reviewing Policy A Conservative:**

Affirmed.

**Discussion Potential:**

4

**Paper Summary:**

This paper argues that the time is right for the AI community to actively propose and promote standards of compatibility in agentic memory systems. The authors claim that the current isolated development landscape of agentic memory systems leads to challenges such as increased integration cost and technical debt as well as the inability to evaluate safety risks and performance within and across systems in fair and comprehensive ways. They believe that instead of continuing down this flexibility-first path or enforcing strict standards, the best solution is to merely incentivize compatibility standards across systems, just like other subfields have done (ML models, containers, Transformers library). After defining and defending the idea of compatibility-first standards, the paper shifts its focus to a more concrete blueprint for efforts in this direction, such as common interfaces, overarching benchmarking efforts and calls for both practitioners and decision makers to value compatibility in their choice of agentic memory systems.

**Position:**

Yes

**Position In Title:**

Yes

**Related Work:**

3

**Strengths And Weaknesses:**

Strengths:
- The position is clearly stated and delineated.
- The evidence and reasoning to support their position is clear.
- The topic is relevant and timely for the language agent community
- The authors go beyond stating a position and propose relatively concrete blueprints for mitigation strategies which will surely encourage discussion.

Weakness:
- Although this position seems rather uncontroversial in the abstract, the main challenges and potential for discussion comes from their implementation. The authors propose some concrete plans in Section 5 but there is little discussion about how there is a lot of work to be done to determine what exact level of `Fragmentation` vs `Standardization` can be currently integrated into an agentic memory interface.

Small Typo:
- Line 350: `Adapter To` in one line

**Support:**

3

---

> ### Author Rebuttal · Authors · 2026-03-31
>
> We thank the reviewer ADFN for their valuable feedback and for providing us with constructive questions/suggestions, which will help us better present this work. We are happy that they found our position paper to be one that provides clear evidence, and consider it a timely and relevant work that encourages discussion. Below we address their concerns, questions and suggestions.
>
> ### W1: Fragmentation vs Standardisation
> We agree that determining the "exact level" of fragmentation vs standardisation is a key question and that it requires further work. At this stage, as the agentic memory systems field is still maturing, we think the best way to approach the compatibility is the provision of a minimal interface, which also matures together with the field. To clarify:
>
> * **Minimal Interfaces**: Our proposal (in Section 5) advocates for core operations (store/retrieve/delete) as the starting point, avoiding over-specification.
>
> * **Community-Driven Evolution**: We think that governance models (e.g., working groups) could iteratively refine standards based on real-world use cases.
>
> * **Future Work**: We will expand on this in the revision, proposing a roadmap for interface extensibility (e.g., adding conflict resolution hooks and semantic update hooks as the field matures).
>
> ### Q1-Q2 on Section Reorganisation
> Your suggestion to group these under "Lack of Evaluation Standards" is insightful and helps avoid any possible confusion. To address this suggestion, in our revision for camera ready version, we:
>
> * Consolidate Sections 2.2. (Safety Risks), 2.3 (Performance) and 2.5 (Evaluation) into a unified section titled "Evaluation and Safety Challenges from Fragmentation.
> * Reframe these as interconnected consequences of incompatible systems (e.g., safety gaps arise when evaluations are siloed).
>
> ### Q3: Memory Conflicts (Section 2.4)
>
> We agree that this is a key point. While memory conflicts and state consistency are going to be one of the main challenges in the next few years for agentic memory systems, regardless of the level of compatibility, we believe that fragmentation significantly exacerbates this issue. Here is our reasoning behind this:
>
> * **Root Cause**: Isolated systems use conflicting state-update semantics (e.g., optimistic vs. pessimistic locking), leading to undetectable inconsistencies when agents interact.
>
> * **Compatibility Solution**: Shared interfaces enable conflict detection/resolution (e.g., via standardised "update" semantics), reducing risks in cross-framework workflows.
>
> * **Example**: An agent using LangGraph's memory might overwrite AutoGen's state if both lack conflict-resolution standards, a pattern that mirrors known challenges in multi-system state management more broadly (e.g.,  prior to adoption of shared transaction protocols, distributed database ecosystem also suffered from memory conflict issues caused by fragmentation [1]. For the agentic memory area, this evidence is still emerging; this Reddit post from the Agentic AI community touches on a related issue [2]).
>
>
> Typo => Thank you for bringing this typo to our attention; we will fix it.
>
> [1] Srinivasan, Jagannathan. "[Replication and fragmentation of composite objects in distributed database systems.](https://dl.acm.org/doi/10.5555/170485)" (1992).
>
> [2] [Memory as infrastructure in multi-agent LangChain / LangGraph systems
> ](https://www.reddit.com/r/LangChain/comments/1rdftbj/memory_as_infrastructure_in_multiagent_langchain/#:~:text=This%20is%20a%20real%20pain,doesn't%20need%20routing%20logic)

---

> > ### Author Rebuttal · Reviewer_ADFN · 2026-04-06
> >
> > Thank you for your response. My concerns are all addressed, however, I would like the authors to include the example used to define "Memory Conflict" in the rebuttal. This could help the readers avoid thinking about "knowledge conflict" in memory systems, which is a completely different problem.
> >
> > I will keep my high score.

---

### Official Review · Reviewer_odSA · 2026-03-17

**Significance:** 4
**Argument Clarity:** 3
**Rating:** 5
**Confidence:** 2

**Questions:**

None

**Alternative Views Section:**

Yes

**Compliance With Llm Reviewing Policy A Conservative:**

Affirmed.

**Discussion Potential:**

4

**Final Justification:**

We thank the authors for their rebuttal. My score adequately reflects the current status of the paper.

**Paper Summary:**

The paper presents an interesting position that agentic memory should be designed for compatibility and not get affected by fragmentation and/or early standardization.

**Position:**

Yes

**Position In Title:**

Yes

**Related Work:**

3

**Strengths And Weaknesses:**

Strengths:
1. The idea of creating compatible agentic memory solutions is a sound position. It will help evaluate alternative ideas and compare solutions in a modular manner.
2. The paper does a good job of investigating alternative ideas such as the natural desire to fragment in such an active area and/or standardization to enforce multiple commercial solutions even at the risk of reduced innovation.
3. Section 5.1 with minimal interfaces and minimal standard interfaces provides a grounded feel to the position paper.

Weaknesses:
1. It is not clear that such minimal interfaces are feasible for agentic memory and if they can truly allow the innovation required for designing novel agentic memory solutions.
2. Fragmentation may be suitable at this early stage; no argument is presented to suggest that compatible agentic memory solutions will not discourage innovation beyond the constraints of the compatible agentic memory solutions.

**Support:**

3

---

> ### Author Rebuttal · Authors · 2026-03-31
>
> We thank the reviewer odSA for their thoughtful response and valuable feedback. We appreciate their interest in our work as a sound position that makes it easier to evaluate and compare different agentic memory solutions. Below we address their concerns, questions and suggestions.
>
> ### W1: Feasibility of Minimal Interfaces
> We acknowledge that in the evolving field of agentic memory systems, whether such minimal interfaces would allow the innovation required for designing new solutions is an open-question and requires further investigation. However, we believe that minimal interfaces (for e.g. retrieval or storage) are designed to be enablers rather than constraints because:
> * **Proven Precedents**: Similarly to Python's database API (DB-API/PEP249) or ONNX's model interoperability, lightweight standards accelerate innovation by reducing integration overhead while allowing internal experimentation (e.g. novel memory architectures such as persistence graph caches).
> * **Extensibility**: The interfaces explicitly support hooks for advanced features (e.g., conflict resolution via an optional `update_semantics` layer), ensuring that they adapt to emerging needs.
>
>
> ### W2: Balancing Compatibility and Innovation
> This is indeed a key issue as fragmentation can be desirable before a field reaches maturity; however, we believe that our compatible-first design proposal provides a sweet spot in balancing these two. Specifically, in our paper, we argue that compatibility prevents fragmentation-induced technical debt, which, in turn, accelerates innovation. This can be best exemplified by historical anecdotes from the previous progress in the ML field:
>
> * **Early-Stage Example**: Rapid evolution of Transformers (from BERT in 2018 to GPT-4 in 2023) thrived partly thanks to shared interfaces (HuggingFace), which removed redundant efforts for reimplementing similar components - a pattern that empirical studies have also subsequently confirmed (Jones et al. [1] find that 80% of pretrained model derivatives adopt the HF Transformers library, indicating consolidation of efforts around a compatible interface).
>
> * **Risk Mitigation**: Our proposal avoids rigid standardisation by:
>   - **Voluntary Adoption**: Frameworks adopt interfaces only if they benefit.
>   - **Governance**: Community-driven updates (e.g., in the form of Request for Comment) ensure interfaces evolve with the field.
>
> [1] Jones, Jason, et al. "[What do we know about Hugging Face? A systematic literature review and quantitative validation of qualitative claims.](https://dl.acm.org/doi/fullHtml/10.1145/3674805.3686665)" Proceedings of the 18th ACM/IEEE international symposium on empirical software engineering and measurement. 2024.

---

> > ### Author Rebuttal · Reviewer_odSA · 2026-04-02
> >
> > We thank the authors for their rebuttal. My score adequately reflects the current status of the paper.

---

### Decision · Program_Chairs · 2026-04-30

**Decision:**

Reject

**Comment:**

The authors argue that some form of compatibility is needed moving forward, following a somewhat traditional software system argument. They draw attention the technical debt, limited reuse of safety tool, poor interoperability, etc. I think the position is timely for the community, and the authors are able to make a credible argument for their position. While I think the paper is good, given the competitive nature of the conference, I wished the authors would have emphasized more the uniqueness of agentic memory, or the ML angle missing, rather than what I see as a traditional software system argument. That is not to say the work is not of value, but unfortunately it just landed right under the acceptance bar.